# Isolation and Evaluation of the Antagonistic Activity of *Cnidium officinale* Rhizosphere Bacteria against Phytopathogenic fungi (*Fusarium solani*)

**DOI:** 10.3390/microorganisms11061555

**Published:** 2023-06-11

**Authors:** Seok Hui Lee, Su Hong Jeon, Jun Young Park, Dae Sol Kim, Ji Ah Kim, Hui Yeong Jeong, Jun Won Kang

**Affiliations:** 1Department of Forestry, Kyungpook National University, Daegu 41566, Republic of Korea; svbreqwaiu@naver.com (S.H.L.); tnghd8802@naver.com (S.H.J.); paul05241@naver.com (J.Y.P.); skyds4499@naver.com (D.S.K.); 2Forest Medicinal Resource Research Center, NIFoS, Yeongju 36040, Republic of Korea; jiahkim@korea.kr (J.A.K.); hdud7281@korea.kr (H.Y.J.)

**Keywords:** *Bacillus*, *Cnidium officinale*, *Fusarium solani*, fusarium wilt, *Leclercia adecarboxylata*, PGPR

## Abstract

*Cnidium officinale Makino*, a perennial crop in the *Umbeliperae* family, is one of Korea’s representative forest medicinal plants. However, the growing area of *C. officinale* has been reduced by plant disease and soil sickness caused by fusarium wilt. This study isolated rhizosphere bacteria from *C. officinale*, and their antagonistic activity was evaluated against *Fusarium solani.* Particularly, four isolated strains, namely, PT1, ST7, ST8, and SP4, showed a significant antagonistic activity against *F. solani*. An in planta test showed that the mortality rates of shoots were significantly low in the PT1-inoculated group. The fresh and dry weights of the inoculated plants were also higher than that of the other groups. The 16S rRNA gene sequencing identified the strain PT1 as *Leclercia adecarboxylata*, and downstream studies confirmed the production of antagonism-related enzymes such as siderophore and N-acetyl-β-glucosaminidase. The phosphorous solubilizing ability and secretion of related enzymes were also analyzed. The results showed that PT1 strain could be utilized as promising plant growth-promoting rhizobacteria (PGPR) and biocontrol agents (BCAs).

## 1. Introduction

*Cnidium officinale Makino* is a perennial crop in the *Umbeliperae* family. It is harvested during the summer and then dried and thinned for a variety of uses such as cosmetics, functional foods, and medicine. *C. officinale* is a representative medicinal plant in Korea, where 80% of its cultivation occurs in the inland Yeongyang county which is in the north-eastern area of North Gyeongsang province. Overall, *C. officinale* is the eighth most cultivated medicinal plant in Korea [1]. The active ingredients of *C. officinale* have been extensively researched. For example, in 2010, an extract from the above-ground part of *C. officinale* was evaluated for its antioxidant properties [1], and in 2011, the anti-inflammatory and anti-cancer effects of its rhizome extract were investigated in in vitro conditions [2].

Currently, there are two main problems with the cultivation of *C. officinale* in Korea, the first of which relates to temperature. Over the past 100 years, the average annual temperature in Korea has risen by 1.5 °C. Furthermore, the Intergovernmental Panel on Climate Change (IPCC) predicts that if humans continue to emit greenhouse gases in line with predictions, the average annual temperature will rise by 5.7 °C in 2100 [3]. However, the optimal temperature for growing *C. officinale* is 8.1–14.0 °C. The production of rhizomes is usually also better at lower temperatures (<14 °C) than at higher temperatures [4]. Thus, if the temperature continues to rise, the areas suitable for *C. officinale* cultivation are predicted to decrease dramatically, and it is expected that only some alpine regions in Gangwon province, in the northeastern part of Korea, will remain after 2090 [5].

The second issue is that *C. officinale* generally has low resistance to infection and can readily be infected by fungi. Major diseases caused by fungi include anthracnose, leaf blight, brown leaf spot, and fusarium wilt. Significantly, *Fusarium* sp., which causes fusarium wilt, can continuously increase with repeated cultivation. However, cultivation techniques to prevent soil sickness that occurs with the repeated cultivation of *C. officinale* have not yet been widely investigated. Therefore, to improve the productivity and sustainability of *C. officinale*, the development of cultivation techniques that can reduce soil sickness and countermeasures that can antagonize pathogens are required [6].

Rhizosphere microorganisms have an essential role in crop physiology. They can be used to control and prevent plant diseases, because their interactions with hosts enhance resistance to pathogens and environmental stresses. Recently, the efficiency and potential of plant growth-promoting rhizobacteria (PGPR) as antagonists and biocontrol agents (BCAs) have been reported in agriculture and forestry. They can compete for nutrients and secrete hormones and antagonists, including indole-3-acetic acid (IAA), strigolactones, siderophore, and protease, against pathogens [7,8].

Therefore, this study aimed to investigate the characteristics of rhizosphere bacteria isolated from the *C. officinale* and evaluate their antagonistic activity against *Fusarium solani*, which is a significant cause of soil sickness.

## 2. Materials and Methods

### 2.1. Isolation of Bacteria from the Rhizosphere and Roots

*C. officinale* seedlings were collected from the fields of farmers in Cheongju, South Korea, and transplanted to pots (60 × 45 × 23 cm; 6 seedlings per pot). After 30 d, three plants that were disease-free and showed good growth were selected, and their rhizosphere soil and roots were sampled. After washing the roots gently, they were then sterilized. First, they were washed with 70% ethanol for 10 s and then 0.5% NaClO for 5 min, and this process was repeated twice. Second, the roots with a sterilized surface were flushed and rinsed three times for 1 min with sterile water. After rinsing the roots, they were crushed with a mortar and pestle. Then, 2 g of crushed roots and 1 g of rhizosphere soil were aliquoted into 50 mL of sterile distilled water, respectively. The mixtures were incubated in a shaking incubator at 26 °C and 150 rpm for 10 min. The mixtures were spread on tryptic soy agar (TSA, MBcell, Seoul, Republic of Korea) and potato dextrose agar (PDA, MBcell, Seoul, Republic of Korea) and further incubated at 26 °C for 2 days. Finally, strains with morphologically different characteristics were isolated in single cultures by streaking on the new same-type medium. The isolated strains were subjected to preliminary dual-culture with *F. solani* and finally selected based on antagonistic activity against *F. solani*.

### 2.2. Identification of Rhizosphere and Root Bacteria

DNA sequences of the selected strains were analyzed by requesting Macrogen (Seoul, Republic of Korea). To identify bacteria, a legion of 16S rRNA was used. DNA sequences were compared to standard databases in the nucleotide database of the National Center for Biotechnology Information (NCBI, Bethesda, MD, USA) using the BLASTn tool. For the phylogenetic tree, DNA sequences were compared to rRNA databases in NCBI using the BLASTn tool [9]. The phylogenetic tree of selected strains was illustrated using a neighbor-joining tree with MEGA 11 v.11, 1000 bootstrap replications, and a maximum composite likelihood substitution model [10].

### 2.3. In Vitro Antagonistic Activity of the Selected Strains

To test the growth inhibition activity of the selected strains against pathogenic fungi, dual-culture with *F. solani* (KACC No. 40384) were performed. The pathogenic fungal strain used in the experiment was obtained from the Korea Agricultural Collection (KACC). Dual-culture were performed in 50 mm diameter Petri dishes with PDA. The pathogenic fungi, in the form of a 4 mm diameter agar plug, were raised on one side, and 3 μL of the overnight cultured strains were inoculated on 6 mm diameter paper disks at a distance of 30 mm from the pathogenic fungi. After incubation at 26 °C for 8 d, the inhibition rate was calculated by measuring the distance between the mycelia of the pathogenic fungi and the selected strains [11]:(1)$$Inhibitionrate(\%)=\frac{Rt}{Rc}\times 100,$$
where *Rc* represents the average growth radius of the mycelia on the control plate and *Rt* represents the distance between the mycelia and bacteria on the treated plate.

### 2.4. In Planta Tests

*C. officinale* rhizomes were divided into 3–4 pieces (8–12 g) based on the buds and then planted in seedbeds (53 × 37 × 9 cm) and germinated for 11 days. Each strain and the pathogenic fungus, *F. solani*, were used to inoculate the treated groups. To check the effects of the infection and the effects of the PGPR for the four strains, untreated seedlings were used as the first control group, and the second control group was inoculated with only *F. solani*. For each group, 16 of the *C. officinale* were used, and 2.5 mL of the selected strains were inoculated at an optical density (OD) of 0.1 at 600 nm, respectively. After incubating for 24 h, the spore suspension of *F. solani* was inoculated into plants at a concentration of 2 × 10^5^ spores/mL, except for the second control group. After 1 week, the selected strains were reinoculated using the same method. Since the damage by the pathogen progressed rapidly and there was a risk that the dead shoots would rot and be lost, the test was terminated after 17 d, and the total number of shoots, dead shoots, and dry weights of shoots and rhizomes were measured. The statistical analysis of the results was performed using one-way analysis of variance and the Statistical Package for the Social Sciences version 25.0, and the post hoc test was performed using the Duncan method.

### 2.5. Temperature Tolerance Assay

To test the growth ability of the strains in relation to temperature, a temperature tolerance assay was conducted [12]. The strains were adjusted to OD 0.01 at 600 nm in 10 mL of TSB medium. Then, these were incubated in a shaking incubator for 150 rpm at 0 °C–40 °C at intervals of 5 °C. For each temperature, the OD value was measured at 600 nm using a UV–Vis spectrophotometer (UBI-490 model, MicroDigital, Seongnam-si, Republc of Korea) at 24 h intervals for 5 d, respectively.

### 2.6. In Vitro Test of PGP Traits

#### 2.6.1. Protease Production Assay

Protease was produced using the PDA medium with 2% skim-milk [13]. Paper disks with 6 mm diameters were placed on the medium, and 3 μL of overnight cultured strains were inoculated. The plate was cultured at 27 °C for 4 d. Clear zones around the paper disk represented the production of protease against protein in the Skim-Milk PDA.

#### 2.6.2. Cellulase Production Assay

Cellulase production was performed using PDA with 1% carboxymethylcellulose (CMC) and 0.01% trypan blue [14]. Paper disks, 6 mm in diameter, were placed on the medium, and 3 μL of overnight cultured strains were inoculated. The plate was cultured at 27 °C for 4 d, and cellulase production was confirmed by a clear zone around the paper disk due to the degradation of CMC combined with trypan blue.

#### 2.6.3. Chitinase Production Assay

The production of chitinase was assessed using PDA with 2% colloidal chitin [15,16]. To make colloidal chitin, 20 g of chitin from shrimp shells was dissolved in HCL by stirring for 2 h. The chitin-HCL solution was filtered to remove undissolved particles using glass wool. After adding 500 mL of distilled water to the solution, the solution was incubated for 24 h at 5 °C, and then the supernatant was removed. To wash the colloidal chitin, 3 L of 5 °C tap-water was added and filtered using Whatman filter paper grade 2. The washed colloidal chitin was suspended in 500 mL of 5 °C distilled water, and the pH was measured. This washing process was repeated until the pH was >7.0. The washed colloidal chitin was stored at 5 °C and used in further experiments. Paper disks with a 6 mm diameter were placed on the medium, and 3 μL of overnight cultured strains were inoculated. After incubating at 27 °C for 4 d. Clear zones around the paper disk represented the production of chitinase against chitin in the colloidal chitin PDA.

#### 2.6.4. Hydrogen Cyanide (HCN) Production

HCN production was assessed using TSA with glycine (4.4 g/L) [17]. The strains were cultured for 24 h and then streaked on medium. The filter papers were immersed in picric acid solution (0.5% picric acid; 1% Na_2_Co_3_) and attached to the upper lids of the Petri dishes and sealed with parafilm. The plates were then incubated at 27 °C for 7 d and HCN production was confirmed based on the color of the filter paper (changes from yellow to red or brown).

#### 2.6.5. Phosphate Solubilization

Phosphate solubilization was assessed in the selected strains using a National Botanical Research Institute phosphate (NBRIP) protocol. The NBRIP medium contained L^−1^; glucose, 10 g; Ca_3_(PO_4_)_2_, 5 g; MgCl_2_·6H_2_O, 5 g; MgSO_4_, 0.14 g; KCl, 0.2 g; (NH_4_)_2_SO_4_, 0.1 g [18]. Paper disks that were 6 mm in diameter were placed on the medium, and 3 μL of the overnight cultured strain was inoculated. The plate was cultured at 27 °C for 4 d, and phosphate solubilizing activity was confirmed by the occurrence of a halo zone around the paper disk.

#### 2.6.6. Siderophore Production

Siderophore production by the selected strains was assayed using a Blue Agar CAS (Chrome azurol S) protocol [19]. Paper disks that were 6 mm in diameter were placed on the medium, and 3 μL of the overnight cultured strains were inoculated. The plates were cultured at 27 °C for 4 days, and siderophore production was confirmed by a halo zone around the paper disk.

#### 2.6.7. Indole-3-Acetic Acid Production

The capacity of indole-3-acetic acid (IAA) production from the selected strains was tested using Salkowski’s reaction [20]. Overnight-cultured strains were adjusted to an OD of 0.1 at 600 nm in 10 mL of tryptophan broth (gelatin pancreatic peptone, 10 g; sodium chloride, 5 g; L-tryptophan, 1 g; pH 7.5). The inoculated broths were cultured in a shaking incubator at 27 °C and 150 rpm for 4 d. Then, 1.5 mL of the cultured broth was centrifuged for 10 min at 7300× *g* at 4 °C, and the supernatant was added to 3 mL of Salkowski reagent (35% HCLO_4_, 98 mL; 0.5 M FeCl_3_, 2 mL) and incubated for 30 min at 25 °C in the dark to allow the reaction to develop. The concentration of IAA was measured using a UV–Vis spectrometer at 530 nm. The standard curve of the IAA to allow quantitative measurements was determined using the following IAA concentrations: 50, 25, 12.5, 6.25, 3.125, 1.563, and 0 mg/mL.

### 2.7. Exoenzyme Activity

Exoenzyme activities were determined using the APIZYM^®^ test system (bioMérieux, Marcy-l’Étoile, France). Overnight cultured strains were adjusted to the turbidity of 6 McFarland in sterile distilled water, and then 65 μL was pipetted to each cupule and incubated at 37 °C for 4 h. Then, ZYM A and ZYM B were added, and the data were recorded and interpreted for 10 min.

## 3. Results

### 3.1. Isolation of Bacteria from the Rhizosphere Soil and Roots

After isolating the strains using PDA and TSA, 7 bacterial strains were isolated from the disease-free and well-growing roots of *C. officinale*, while 21 bacterial strains were isolated from the rhizosphere soil. After dual-culture with *F. solani*, four of the bacterial strains showed high levels of inhibition activity and were selected for further investigation. The selected strains were identified using their 16S rRNA genes, and the BLASTn results for the sequences from the GenBank database are summarized in Table 1 and Figure 1.

### 3.2. In Vitro Antagonistic Activity of the Selected Strains

The dual culture test to assess the antagonistic activity of the selected strains against *F. solani*, showed that all four strains were confirmed to have inhibitory activity, as judged by the occurrence of the inhibition zone. Among them, the PT1 showed the highest inhibition rate at 21.1 ± 2.4%, and this was followed by ST7 at 16.3 ± 4.9%, ST8 at 11.5% ± 1.4%, and SP4 at 10.9 ± 3.1% (Figure 2).

### 3.3. In Planta Tests

Each of the strains and *F. solani* were inoculated into *C. officinale* and monitored for 18 d. After 17 d, the number of shoots, number of dead shoots, and the shoot fresh and dry weights were measured. An average of 17.8 new shoots was found to be sprouted from each plant, and the average shoot death rate in the first control group was 28.4%. In the treated and second control groups, there was an average of 47.6% shoot death rate, except with PT1 treated groups, which had an average death rate of 27% that was similar to the first control group (Figure 3b). Furthermore, the dry weights of PT1 group were highest among all groups (Figure 3c,d). However, the standard deviation seemed to be largest in the PT1 group, as some plants were suppressed by the surrounding well-grown plants or had minor responses to the inoculation.

### 3.4. Growth Traits in Response to Temperature

OD measurements were taken every 24 h for 5 days. Above 35 °C, all strains showed slow growth rates and PT1 did not grow (Figure 4d). However, at 10–30, PT1 and ST7 showed the highest growth rates among all strains and, at temperatures < 5 °C, only PT1 and ST7 were shown to have growth (Figure 4a–c). All strains achieved their highest growth rate at 15 °C (Figure 4b).

### 3.5. In Vitro Test of PGP Traits

As a result of the in vitro test of plant growth-promoting (PGP) traits according to each assay, the ST8 and SP4 showed production of protease and cellulase related to antifungal ability by degrading the cell membranes and cell walls of fungi, respectively. The PT1 and ST7 showed production of siderophore related to antifungal ability and Fe uptake in plants, and capacity of phosphate solubilizing related to uptake of P in plants. In the test of IAA production, a plant growth-promoting hormone, only PT1 secreted 13.19 ± 4.22 ppm of IAA. However, there were no strains producing chitinase and HCN (Figure 5 and Table 2).

Each IAA value represents the mean of three replicates and the standard deviation (SD).

### 3.6. Exoenzyme Activity

The exoenzyme characterization of strain enzymes using APIZYM confirmed that seven types of enzymes (alkaline phosphatase, esterase, esterase lipase, acid phosphatase, naphthol-AS-BI-phosphohydrolase, α-glucosidase, and β-glucosidase) were mainly produced in all strains. α-glucosidase was produced by SP4 and ST8. β-glucosidase was produced by ST7 and ST8. However, PT1 produced N-acetyl-β-glucosaminidase instead of glucosidase (Table 3).

## 4. Discussion

In this study, we isolated and evaluated bacteria from the roots and rhizosphere of *C. officinale*, then further screened strains based on their antagonistic ability against *F. solani*, which causes fusarium wilt [21,22]. Four strains have been identified as antagonistic bacteria. Among these, only the group with PT1 treatment showed a significant effect with the in planta test. In the other groups, the mortality rates of the shoots increased after being treated with pathogenic fungi compared with the first control group. However, in the group treated with PT1, the mortality rate of the shoots was not significantly different when compared with the first control group, while the fresh and dry weights increased. In addition, the PT1 treated group had the highest inhibition rate in the in vitro antagonistic activity test. The PT1 strain was thus considered effective at controlling fusarium wilt in *C. officinale*.

Temperature is an important factor in the growth of plants and bacteria. *C. officinale* grows well in cool climates in the semi-highlands, which are approximately 400 m above sea level and have a wide diurnal range [4]. Yeongyang county is the largest production region for *C. officinale*, and its annual average temperature is 9–10 °C, the average temperature from April–October during the cultivation period is 15–17 °C, and the maximum temperature does not exceed 28 °C [23]. The results of this study have shown that the PT1 strain can grow in temperatures ranging from 5 °C to 30 °C, which may be broader than the range required for *C. officinale.* Therefore, the application of PT1 to the actual cultivation region is considered feasible.

In the APIZYM test, the PT1 was confirmed to produce acid phosphatase and naphthol-AS-BI-phosphohydrolase related to phosphate solubilization [24,25], in addition, N-acetyl-β-glucosaminidase and several hydrolases were produced, and high siderophore production ability was confirmed using the CAS medium test. It is thus considered that PT1 has the ability to solubilize and convert insoluble phosphorous and iron in the soil into forms usable by plants, respectively [26,27,28]. In particular, PT1 strain showed outstanding phosphorus solubilizing activity when compared with the other selected strains. In addition to solubilizing iron in the soil, siderophores are also related to the bacterial ability to antagonize pathogenic fungi [29,30,31], and N-acetyl-β-glucosaminidase can damage the cell wall of fungi and inhibit their growth [32,33].

The PT1 strain was specifically identified as *Leclercia adecarboxylata*. It is a gram-negative anaerobic bacillus [34,35], and it was first reported by Leclerc in 1962 [36]. This species has previously been isolated from herbaceous crops and investigated due to its ability to secrete IAA and biomolecules to increase resistance to abiotic stress or promote plant growth [37,38]. Moreover, the existence of antifungal activity against phytopathogenic fungi and gene-related root colonization capacity has been previously studied [37,39,40].

Fusarium wilt occurs after the penetration of mycelium or germinating spores into the plant root tissues, from which they enter the xylem and produce microconidia [41,42]. Therefore, to control fusarium wilt, it is important for bacteria to penetrate the plant root tissues and antagonize pathogenic fungi through plant–microbe interactions. The PT1 strain (*L. adecarboxylata*) was isolated from the root tissues and was previously found to be capable of root colonization. Thus, it is expected that the PT1 strain can penetrate the root tissue and inhibit the growth of pathogenic fungi, and the traits of PT1 have high potential to be developed as a BCA in the future.

The other three strains identified were *Bacillus subtilis*, *Bacillus inaquosorum*, and *Bacillus vallismortis* [43,44,45]. Previous studies have also reported that these strains have antifungal activity [46,47,48]. Although these strains did not show significant effects in the planta test, they were confirmed to produce glucosidase and protease unlike the PT1. Glucosidase catalyzes the hydrolysis of starch into a simple form [49], and recent studies have reported that it can protect plants against pathogen [50]. Protease can also antagonize pathogens and is an important enzyme in the nitrogen cycle [51,52].

The *Bacillus* sp. is major bacterium in the rhizosphere, and it can survive in the soil for a long period under harsh environmental conditions via spore formation [53]. In particular, *Bacillus subtilis* had been studied for its potential as a PGPR [54], and it has been reported that root length and dry weight significantly increased when inoculated into the rhizosphere of maize [55]. Various microorganisms are involved in plant growth, and isolating and utilizing various microbial resources is also beneficial to the stability of the soil ecosystem [56]. Previous studies have reported that inoculations with two or more strain mixtures had better growth-promoting effects than inoculations with only one strain [57]. Therefore, this study isolated various PGPRs involved in promoting the growth of *C. officinale* and suggested the possibility of using them as biocontrol agents in the future.

## 5. Conclusions

The results of this study showed that both the PT1- and pathogenic fungi-inoculated group had higher growth rates of plants than the untreated control group. An in vitro test revealed that the strain PT1 secreted hormones and enzymes that had antifungal and plant growth-promoting effects. Thus, the PT1 strain was considered a promising PGPR and BCA. It can be used to counteract the issue of climate warming, because the optimum range of the growth temperature of PT1 is broader than that of *C. officinale*. However, the optimization of inoculating techniques in terms of methods, concentrations, inoculation frequency, and timings should be investigated further. A field test should also be conducted to evaluate its effectiveness [58]. Therefore, the PT1 strain can be considered a promising PGPR and BCA.

## Figures and Tables

**Figure 1 microorganisms-11-01555-f001:**
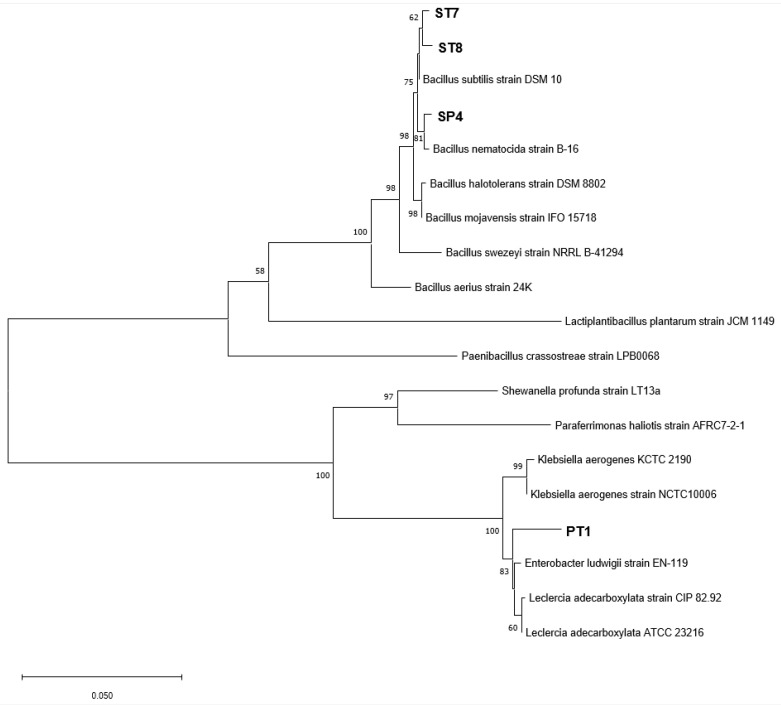
Neighbor-joining phylogenetic tree based on the 16S rRNA genes of each selected strain. The trees were compared to the NCBI 16s rRNA database. The numbers at the nodes indicate bootstrap values >50% with 1000 bootstrap replications. ST7, ST8, and SP4 were isolated from rhizosphere soil and PT1 was isolated from the root tissue. *Lactiplantibacillus* sp., *Paenibacillus* sp., *shewanella* sp., and *Paraferrimonas* sp. are used as the out-group.

**Figure 2 microorganisms-11-01555-f002:**
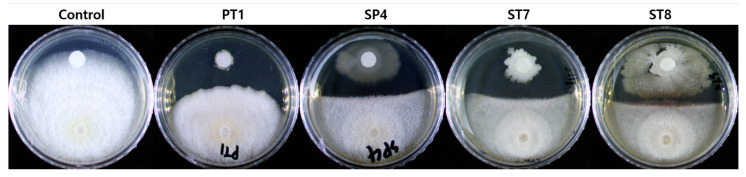
Dual culture test for each of the selected strains to show antagonistic activity against *F. solani* (pathogenic fungi). The top half of each plate is inoculated with the selected strain on a paper disk, and the lower half is inoculated with *F. solani* in the form of an agar plug.

**Figure 3 microorganisms-11-01555-f003:**
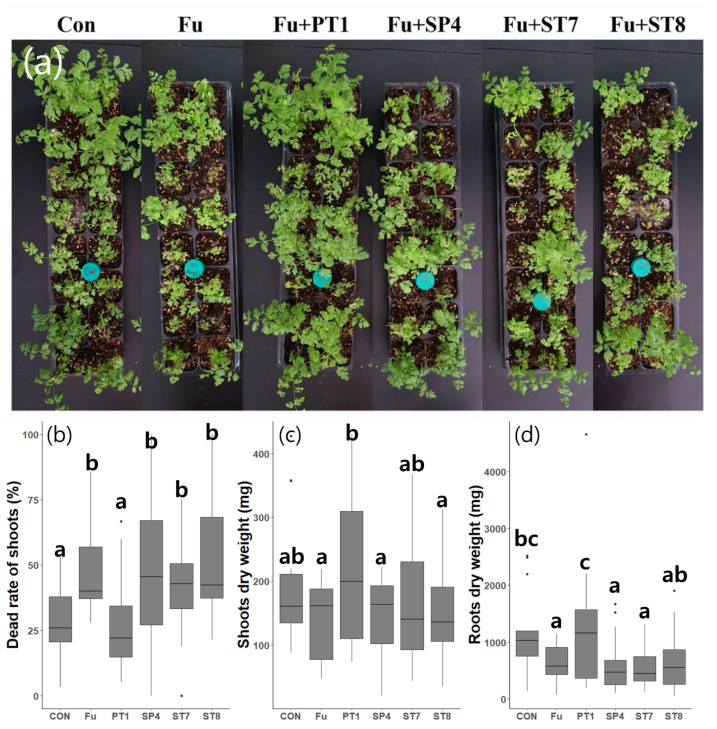
In planta test for the antagonistic activity of each selected strain. (**a**) *C. officinale* were maintained at 25 °C for 17 d after inoculation (after 28 d of growth); boxplots showing rate of (**b**) dead shoots; (**c**) shoot dry weight; and (**d**) root + rhizome dry weight. Different letters indicate statistically significant differences (*p* < 0.05). Con, first control group that was not inoculated; Fu, a second control group was inoculated with only *F. solani* (pathogenic fungi); PT1, SP4, ST7, and ST8 groups were inoculated with each of the selected strains and *F. solani*, respectively. Line in the box represents median, and the top and bottom of the box represent the 75th and 25th percentile, respectively. Whiskers represent minimum and maximum values and points represent outliers.

**Figure 4 microorganisms-11-01555-f004:**
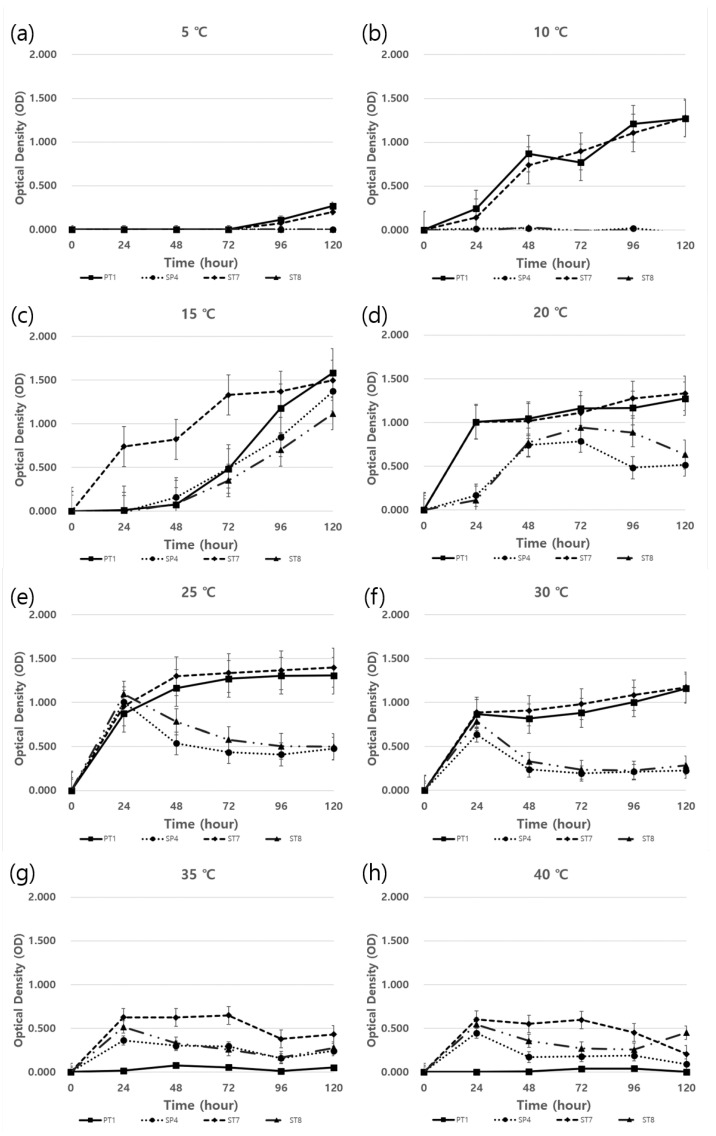
Bacterial growth curve characterization for each selected strain under different temperatures (5–40 °C): (**a**) 5 °C; (**b**) 10 °C; (**c**) 15 °C; (**d**) 20 °C; (**e**) 25 °C; (**f**) 30 °C; (**g**) 35 °C and (**h**) 40 °C. Each strain was inoculated in TSB broth at OD 0.01 at 600 nm and cultured for 120 h. Each point represents the mean of three replicates and stand error (SE).

**Figure 5 microorganisms-11-01555-f005:**
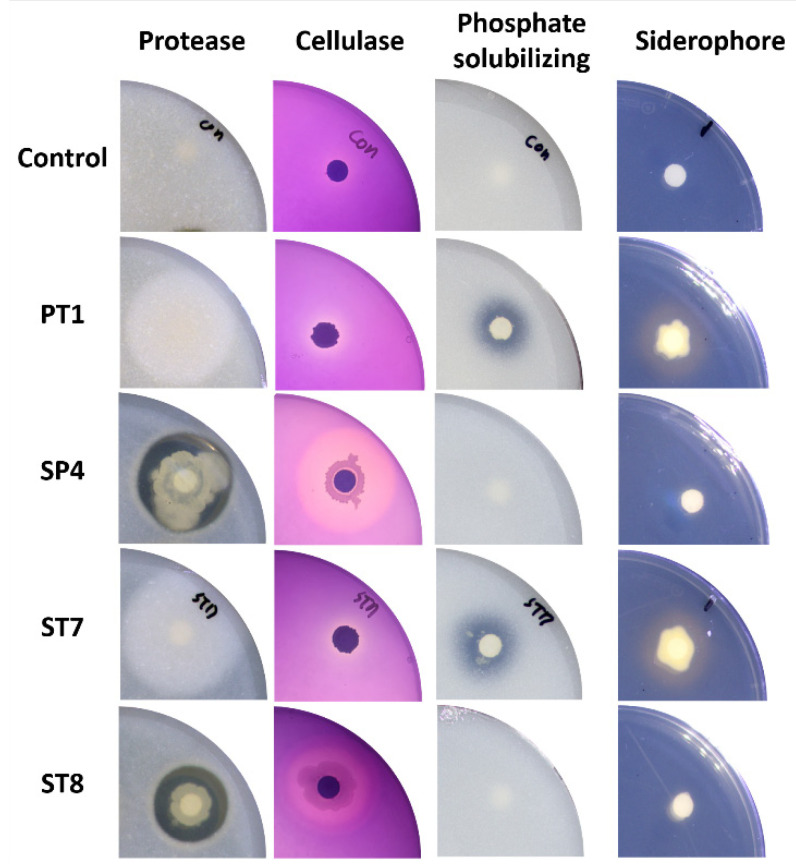
Agar plates showing the production of various enzymes for each selected strain. For the protease production, cellulase production, and phosphate solubilizing tests, strains that formed a clear zone indicated a positive result. For the siderophore production test, a strain that formed a halo zone indicated a positive result.

**Table 1 microorganisms-11-01555-t001:** BLASTn results for the four selected strains in the standard NCBI database.

Strains	16S rRNA Identification ^1^	GenBank Accession No. ^1^	Similarity (%)	Isolated Sources/Medium
ST7	*Bacillus inaquosorum*	NR_104873.1	99	Rhizosphere soil/TSA ^2^
ST8	*Bacillus subtilis*	CP019663.1	99	Rhizosphere soil/TSA ^2^
SP4	*Bacillus vallismortis*	NR_024696.1	99	Rhizosphere soil/PDA ^3^
PT1	*Leclercia adecarboxylata*	NR_104933.1	99	Root tissue/TSA ^2^

^1^ Species and strains from each database with the highest levels of similarity are indicated; ^2^ TSA: tryptic soy agar; ^3^ PDA: potato dextrose agar.

**Table 2 microorganisms-11-01555-t002:** Enzyme production characterization to assess the PGP traits of each strain.

Strain	Protease	Cellulase	Chitinase	HCN	PS	SP	IAA (ppm)
PT1	−	−	−	−	+	+	13.19 ± 4.22
SP4	+	+	−	−	−	−	NA
ST7	−	−	−	−	+	+	NA
ST8	+	+	−	−	−	−	NA

+ indicates a positive result and − indicates a negative result. HCN, hydrogen cyanide; PS, phosphate solubilizing; SP, siderophore; IAA, indole-3-acetic acid.

**Table 3 microorganisms-11-01555-t003:** Enzyme production characterization of each strain using APIZYM.

Enzyme	PT1	SP4	ST7	ST8
Alkaline phosphatase	+	+	+	+
Esterase (C4)	+	+	+	+
Esterase lipase (C8)	+	+	+	+
Lipase (C14)	−	−	−	−
Leucine arylamidase	−	−	−	−
Valine arylamidase	−	−	−	−
Crystine arylamidase	−	−	−	−
Trypsin	−	−	−	−
a-chymotrypsin	−	−	−	−
Acid phospatase	+	−	+	+
Naphthol-AS-BI-phosphohydrolase	+	+	+	+
α-galactosidase	−	−	−	−
β-glucuronidase	−	−	−	−
β-glucosidase	−	−	−	−
α-glucosidase	−	+	−	+
β-glucosidase	−	−	+	+
N-acetyl-β-glucosaminidase	+	−	−	−
α-mannosidase	−	−	−	−
α-fucosidase	−	−	−	−

+ indicates a positive result and − indicates a negative result.

## Data Availability

All data related to this manuscript are incorporated in the manuscript only.

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
