# Peer review of "Isolation and Evaluation of the Antagonistic Activity of Cnidium officinale Rhizosphere Bacteria against Phytopathogenic fungi (Fusarium solani)"

_microorganisms, 2023, doi:10.3390/microorganisms11061555_

Round 1
Reviewer 1 Report
The authors of the manuscript “Isolation and evaluation of the antagonistic activity of Cinidium officinale rhizosphere bacteria against phytopathogenic fungi (Fusarium solani)” addressed an attractive scientific topic important from the point of view of the sustainable approaches’ development in terms of plant protection as one the biggest challenges in agricultural production.
Overall, it is a well-written manuscript, with relevant information, detailed methodology, and a lot of results. Still, it is necessary to improve some segments of the manuscript and the interpretation of obtained results. The authors made a remarkable contribution to the research literature in this investigation field, and with the mentioned concerns corrections, the manuscript could be accepted for publication.
· To improve the narrative approach in the manuscript, it is best to avoid personal pronouns in a scientific writing style.
· In addition to the fact it is a well-described approach in the literature, the introduction section should still include more information on biocontrol as a plant pathogens management approach, by describing the possible mechanisms of action, and simultaneously comparing it with the current methods applied for pathogens suppression. Also, the introduction should include a more detailed explanation of the PGPR abilities and beneficial activities contributing to agricultural production.
· The final paragraph of the introduction should state the hypothesis by indicating the missing information in the literature, and the focus of the conducted study.
· Emphasize the perspectives of future work and the following research steps, focusing also on the contribution of the conducted research in terms of potential application in agricultural practice and replacement of conventionally used methods, in the discussion section as well as in the concluding remarks.
Minor editing of English language required.
Author Response
Point 1: it is necessary to improve some segments of the manuscript and the interpretation of obtained results. The authors made a remarkable contribution to the research literature in this investigation field, and with the mentioned concerns corrections, the manuscript could be accepted for publication.
Response 1: Thank you very much for your comments and suggestions. To improve the manuscript, we have made relevant changes per your comments, and the revised manuscript has been proofread by an English native speaker.
Point 2: To improve the narrative approach in the manuscript, it is best to avoid personal pronouns in a scientific writing style.
Response 2: We have removed and avoided the use of personal pronouns in the manuscript.
Point 3: In addition to the fact it is a well-described approach in the literature, the introduction section should still include more information on biocontrol as a plant pathogens management approach, by describing the possible mechanisms of action, and simultaneously comparing it with the current methods applied for pathogens suppression. Also, the introduction should include a more detailed explanation of the PGPR abilities and beneficial activities contributing to agricultural production.
Response 3: We have added possible mechanisms of PGPR in lines 57–61. The added text is as follows: “Recently, the efficiency and potential of plant growth-promoting rhizobacteria (PGPR) as antagonists and biocontrol agents (BCAs) have been reported in agriculture and forestry. They can compete for nutrients and secrete hormones and antagonists including indole-3-acetic acid (IAA), strigolactones, siderophore, and protease against pathogens.”
Point 4: The final paragraph of the introduction should state the hypothesis by indicating the missing information in the literature, and the focus of the conducted study.
Response 4: We have revised and added some information in lines 62–64. The added information is as follows: “Therefore, this study aimed to investigate the characteristics of rhizosphere bacteria isolated from the C. officinale and evaluate their antagonistic activity against Fusarium solani, which is a significant cause of soil sickness.”
Point 5: Emphasize the perspectives of future work and the following research steps, focusing also on the contribution of the conducted research in terms of potential application in agricultural practice and replacement of conventionally used methods, in the discussion section as well as in the concluding remarks.
Response 5: We have added some sentences to describe the prospective future work and research steps in the discussion and conclusion sections in lines 356 – 363
“It can be used to counteract the issue of climate warming, because the optimum range of the growth temperature of PT1 is broader than that of C. officinale. However, optimization of inoculating techniques such as methods, concentrations, inoculation frequency, and timings should be investigated further. A field test should also be conducted to evaluate its effectiveness [58]. Therefore, PT1 strain can be considered a promising PGPR and BCAs.”

Reviewer 2 Report
In this manuscript, the authors describe the isolation, enzymatic activities and antagonistic activity of Cnidium officinale rhizosphere bacteria against the phytopathogenic fungus Fusarium solani.
The paper is clear and interesting, although it needs to be checked by an English native speaker
Minor comments:
In the title the authors write Cinidium officinale while in the rest of the paper they write Cnidium officinale, please correct the name of the crop reported in the title
- In the abstract please specify what some abbreviations mean, such as PT1 and PGPR reported for the first time
- 2.1 line 75, please add the bibliography concerning the composition or the manufacturers of TSA and PDA media
Finally, authors should report whether the antagonistic activity of the isolated strains is comparable to that of strains already known as antagonists of phytopathogenic fungi
This manuscript needs to be checked by an English native speaker.
Here, as an example, two sentences to correct
Discussion, lines 327-337 The Bacillus sp. is the major bacteria in the rhizosphere, and they can survive....... ....
Therefore this study isolated various PGPR involved in promoting the growth of C.officinale and suggest ...
Author Response
Point 1: The paper is clear and interesting, although it needs to be checked by an English native speaker
Response 1: Thank you very much for your comments and suggestions. As suggested, the revised manuscript has been proofread by an English native speaker.
Point 2: In the title the authors write Cinidium officinale while in the rest of the paper they write Cnidium officinale, please correct the name of the crop reported in the title
Response 2: We apologize for the typographical error. “Cinidium officinale” has been replaced with “Cnidium officinale.”
Point 3: In the abstract please specify what some abbreviations mean, such as PT1 and PGPR reported for the first time
Response 3: We explained the abbreviation “PGPR” as “plant growth-promoting rhizobacteria (PGPR)” and “biocontrol agents (BCAs) in lines 23–24.
PT1, ST7, ST8, and SP4 are strain names and have no definitions. PT strains were isolated from Plant tissue and TSA medium, and SP were isolated from Soil and PDA medium. It is difficult to explain the same in the abstract. We added the source of isolation in Table1 (lines 208–209).
Point 4: 2.1 line 75, please add the bibliography concerning the composition or the manufacturers of TSA and PDA media
Response 4: We added the bibliography after TSA and PDA in line 83–84, “(TSA, MBcell, Seoul, Korea)” and “(PDA, MBcell, Seoul, Korea)”
Point 5: authors should report whether the antagonistic activity of the isolated strains is comparable to that of strains already known as antagonists of phytopathogenic fungi
Response 5: We added the following sentences and references based on your suggestion.
“Moreover, the existence of antifungal activity against phytopathogenic fungi and gene-related PGPR and root colonization capacity has been studied [37, 39, 40].” in discussion section (line 323 - 324).
“Previous studies have also reported that these strains have antifungal activity [45, 46, 47]” in line 334 - 335 to report each strain is comparable to already known strains.
